# Heart Rate Variability in Psychology: A Review of HRV Indices and an Analysis Tutorial

**DOI:** 10.3390/s21123998

**Published:** 2021-06-09

**Authors:** Tam Pham, Zen Juen Lau, S. H. Annabel Chen, Dominique Makowski

**Affiliations:** 1School of Social Sciences, Nanyang Technological University, Singapore 639818, Singapore; phamttam17@gmail.com (T.P.); lauzenjuen@gmail.com (Z.J.L.); dom.makowski@gmail.com (D.M.); 2Centre for Research and Development in Learning, Nanyang Technological University, Singapore 637460, Singapore; 3Lee Kong Chian School of Medicine, Nanyang Technological University, Singapore 636921, Singapore; 4National Institute of Education, Nanyang Technological University, Singapore 637616, Singapore

**Keywords:** HRV, ECG, respiration, biosignals, psychophysiology, psychology, NeuroKit2

## Abstract

The use of heart rate variability (HRV) in research has been greatly popularized over the past decades due to the ease and affordability of HRV collection, coupled with its clinical relevance and significant relationships with psychophysiological constructs and psychopathological disorders. Despite the wide use of electrocardiograms (ECG) in research and advancements in sensor technology, the analytical approach and steps applied to obtain HRV measures can be seen as complex. Thus, this poses a challenge to users who may not have the adequate background knowledge to obtain the HRV indices reliably. To maximize the impact of HRV-related research and its reproducibility, parallel advances in users’ understanding of the indices and the standardization of analysis pipelines in its utility will be crucial. This paper addresses this gap and aims to provide an overview of the most up-to-date and commonly used HRV indices, as well as common research areas in which these indices have proven to be very useful, particularly in psychology. In addition, we also provide a step-by-step guide on how to perform HRV analysis using an integrative neurophysiological toolkit, NeuroKit2.

## 1. Heart Rate Variability for Psychologists: A Review of HRV Metrics and an Analysis Tutorial

A complex and constantly changing heart rate (HR) is an indicator of healthy regulatory systems that can effectively adapt to sudden environmental and psychological challenges [1,2,3,4,5]. Thus, reduced heart rate variability (HRV) has not only been associated with poor cardiovascular health outcomes [6,7,8] and a range of vascular diseases [4,9,10], but also with different mental disorders and cognitive impairments [8,11,12,13,14,15,16,17].

The physiological mechanism underlying HRV exemplifies the tight, yet complex coupling between the brain and the body. Heartbeats are initiated by the impulse generated from the sinoatrial (SA) node—a bundle of specialized cells in the heart. The SA node, dictating the rhythm of heart contractions, is in turn controlled by the complex interactions between different regulatory systems and mechanical activities such as respiration. As the interdependent sources of HR operate on different time scales, HRV is itself unevenly distributed in different frequency bands, with high and low frequency changes corresponding to different neurophysiological mechanisms. Rapid changes of HR reflect the cardiac regulatory influences of the autonomic nervous system (ANS), in tandem with its dynamic interaction with cardiovascular and respiratory activities [10,18,19,20]. On the other hand, slow HR changes reflect bodily fluctuations of lower frequency, such as circadian rhythms, sleep cycles, metabolism, and changes in body temperature and hormonal systems.

The autonomic nervous system (ANS), which influences HR on short-time intervals, is subdivided into two distinct components, namely the sympathetic (SNS) and the parasympathetic nervous system (PNS). The SNS, known as the “quick response” system, predominates during elevated activity and stressful states and its stimulation results in an increase in HR. The PNS, known as the “relaxed response” system, predominates in the quiet and relaxing states, and is related to a decrease in HR. In healthy individuals, the two systems work together (promptly alternating between each other) to maintain the regulatory balance in physiological autonomic function.

In addition to the neural contributions of the two ANS systems, the short-term periodic changes of HR are also influenced by the cyclic activation of the baroreceptors—a mechanical sensor that works to maintain a homeostatic level of blood pressure [21]. When there is an increase in blood pressure, baroreceptors fire to decrease HR and increase the diameter of blood vessels, which subsequently brings the blood pressure back to normal. The opposite effect is observed when the blood pressure decreases. This ability of the body to regulate blood pressure is referred to as baroreflex. The stronger the baroreflex, the more frequently our HR is adjusted to maintain a stable level of blood pressure. Thus, a stronger baroreflex is associated with greater HRV.

The baroreflex is also the autonomic reflex that underlies the physiologic interactions between the cardiovascular and respiratory systems. Specifically, the arterial baroreceptors are also affected by breathing (arterial pressure decreases during inhalation and increases during exhalation), which results in variation in HR during each breathing cycle. This specific influence of respiration on HRV is referred to as respiratory sinus arrhythmia (RSA). RSA can be seen as one of the aspects of HRV and encapsulates the coupling between respiratory and cardiac activity. It has been measured via the proxy of specific HRV indices sensitive to the variability in the respiratory frequency band (e.g., in [22]). However, RSA-specific metrics, which usually require the respiratory signal to quantify selectively the variability related to breathing, provide a more accurate measure of RSA [23].

Due to the different time scales and frequencies at which different systems influence HR, the duration of the cardiac activity recordings naturally limits the sources of variability that can be quantified. While short recordings of 5 min or less can effectively capture high-frequency HRV, long-term recordings of at least 24 h are needed to reliably assess lower-frequency components [18,20]. As long-term recordings capture changes across more diverse situations, they are more reflective of the general state of the system and have been found to be more predictive of health outcomes [24,25,26,27]. However, due to practical considerations, there is a strong interest regarding the reliability (and validity) of indices computed from shorter recordings.

Recent evidence suggests that some HRV indices obtained with time windows as short as 10–30 s are as reliable as short-term HRV indices obtained from more typical (i.e., 5 min or more) recording lengths [28,29,30,31,32,33]. However, this only applies to some “types” of HRV metrics, namely time-domain indices, rather than frequency-domain indices (which require at least 1 min) [33,34] or non-linear indices. Other metrics, specifically targeting long-term changes, can require as long as 24 h of recording to capture the mechanisms of interest. We will discuss these different types of HRV indices in the following section.

## 2. HRV Indices

When HRV was first recognized as a potential indicator of ANS abnormalities [35], the main method used to quantify it was to measure the amount of variance in inter-beat intervals (IBI)—the time intervals between successive heartbeats. In 1981, Akselrod et al. [36] introduced the power spectral analysis of HR to assess the power (or energy) found within different frequency bands. As different regulatory systems influence HRV at different frequencies, frequency-domain measures allow the assessment of their unique contributions to changes of HR. A few years later, L. Cohen [37] introduced the time-frequency domain measures to capture both the time and frequency variabilities simultaneously. In the same decade, Goldberger and West [38] argued that the HR signal was not simply a product of a regular periodic oscillator; instead, it displayed characteristics of a deterministic chaos—a seemingly random sequence of events with an underlying structure [39,40]. Therefore, in order to capture the complex non-linear dynamic behaviors of the “chaotic” fluctuations of HR, new analytical approaches based on entropy and fractal dimensions were proposed [40,41,42,43]. Since then, the evolution of HRV research has been associated with the emergence of new and supposedly better indices or approaches to quantify and reliably assess HRV, making it a rapidly evolving field heavily connected to mathematical and computational developments.

## 3. Time-Domain Analysis

Time-domain measures reflect the total variability of HR and are relatively indiscriminate when it comes to precisely quantifying the respective contributions of different underlying regulatory mechanisms [44]. However, this “general” sensitivity can be seen as a positive feature (e.g., in exploratory studies or when specific underlying neurophysiological mechanisms are not the focus). Moreover, as they are easy to compute and interpret, time-domain measures are still among the most commonly reported HRV indices.

## 4. Deviation-Based Indices

*SDNN*, measured in the unit of milliseconds (ms), is the standard deviation of the normal beat intervals (denoted NN intervals), i.e., excluding beats related to technical artefacts and physiological artefacts (e.g., similar to ectopic beats—beats that do not arise from SA node depolarizations—and atrial fibrillation). Mathematically, *SDNN* is the square root of the total variance in the entire recording and, therefore, it encompasses both short-term high frequency variation (mostly due to parasympathetically-mediated RSA [20]) and long-term low frequency components of the HR signals. Some proponents argue that the accuracy of *SDNN* is higher for longer recording periods, especially over 24 h periods where values of *SDNN* have been used for cardiac risk stratification in medical settings [20].

Stemming from the concept of SDNN, there are other statistical variables that similarly capture the variance of NN but in a more fine-grained manner. Firstly, the standard deviation of average NN is extracted for each 5 min segment of the long-term HR time series (e.g., 24 h recordings) (*SDANN*). It is highly correlated with *SDNN* and has been said to provide little additional information beyond what *SDNN* is able to [20]. On the other hand, the SDNN index (*SDNNI*) is the mean of the standard deviations of NN calculated per 5 min segment [20]. While *SDANN* is an estimate of long-term components contributing to HRV, *SDNNI* assesses the variability within the short intervals and hence is an estimate of short-term components that modulate HR signals.

While these traditional indices have been recommended by official guidelines [18] and have been extensively used in past research, new metrics are being explored in parallel with the development of the mathematics and statistics fields. Some of them involve certain forms of normalization, such as the Coefficient of Variation of NN (CV), i.e., the ratio of the standard deviation to the mean, which could be more stable when comparing individuals or conditions with different HR baselines [45]. Others involve “robust” alternatives to traditional dispersion indices, such as the Median Absolute Deviation (MAD; a median-based index of variability) or the interquartile range (IQR), that are less influenced by outliers and, thus, less sensitive to measurement errors [46,47]. 

## 5. Difference-Based Indices

While the aforementioned indices are calculated directly from the NN sequence, other approaches are derived from the difference between successive NN intervals. These measures include the standard deviation of the successive difference (*SDSD*), the root mean square of successive difference (*RMSSD*), the proportion of successive NN intervals that are larger than a given threshold (e.g., 20 ms or 50 ms—*pNN20* and *pNN50*, respectively). As these measures are derived from beat-to-beat differences, they are mostly indicative of short-term variations in HR.

Among the highly correlated difference-based indices, *RMSSD* is generally preferred by researchers due to its better statistical properties [18,48]. As compared to *SDNN*, *RMSSD* is more influenced by the PNS activity and hence is often used to estimate the vagally mediated fluctuations in HR [19,48]). Its statistical robustness makes it well-suited for short-term time windows [30,33,49].

## 6. Geometric Indices

HRV can also be quantified from geometric (i.e., graphical) representations of NN, in the form of histograms and scatter plots. Such indices include, among others, the HRV triangular index (*HTI*), the triangular interpolation of NN histogram (*TTIN*) and other geometrical properties of the NN scatter plots (e.g., Lorenz plots). While geometric methods are less sensitive to measurement errors as compared to basic dispersion-based indices [50], a large number of NN intervals is required for reliable assessment. Therefore, even though 5 min recordings have successfully been used in past research [51], recordings of at least 20 min are recommended for accurate computation [18].

## 7. Frequency-Domain Analysis

As the different regulatory systems modulate HR at distinct frequencies, frequency-domain indices that reflect the distribution of power across different frequencies bands are particularly suited to assess the specific components of HRV. According to Force [18], the HRV power spectrum can be meaningfully divided into four bands: ultra-low frequency (*ULF*; ≤0.003 Hz), very low frequency (*VLF*; 0.0033–0.04 Hz), low frequency (*LF*; 0.04–0.15 Hz) and high frequency (*HF*; 0.15–0.4 Hz).

Generally, due to the inherent differences in their signaling mechanisms, the parasympathetic (vagal) modulation of HR, including RSA, is faster than the sympathetic activities. Thus, research suggests that *HF* and *LF* are prominent reflections of parasympathetic and sympathetic activity, respectively [44,52]. The ratio of LF to HF power (*LF/HF*) has also been commonly used as an index of the sympathovagal balance—a measure of the relative contributions of SNS to PNS activity.

*HF* component is relatively established as an index of the parasympathetic modulation of HR and under controlled conditions, its derivative, *LnHF* (the natural logarithm of HF) as an estimator of vagal tone [20,44]. However, the concept of tagging a frequency component to a particular division of ANS activities has been controversial for other components. Particularly, the interpretation of LF power as an index of sympathetic activities has been repeatedly challenged. In contrast to the assumption that LF power dominantly represents the sympathetic component, research has shown that it is modulated by both ANS branches as well as baroreceptor activities [18,44,53,54]. There are also mixed findings regarding the exact physiological mechanisms underlying *ULF* and *VLF* components [20,27], with the former often being associated with very low-frequency biological processes such as circadian rhythms or metabolisms [55,56] and the latter with body temperature regulation and vasomotor activity [57,58,59].

Another limitation of the frequency-domain approach is pertaining to the inherent differences in the outputs of different spectral analysis algorithms, which make it difficult to compare the absolute values of frequency components across studies [18,60]. For instance, a non-parametric method such as the fast Fourier transform (FFT) periodogram has been argued to overestimate the *LF/HF* as compared to the Lomb–Scargle (LS) periodogram [60]. Even with the same algorithm, the values of the frequency components can greatly vary depending on the choices of the parameters such as the length of the estimation window or the model order [61].

To allow for more reliable comparison of frequency components across studies, the spectral indices are often normalized by dividing the individual component by the total power. The normalization process allows the normalized components (normalized LF, *LFn* and normalized HF, *HFn*) to be less varied by the methodological differences and less sensitive to the changes in total power [18,62]. Depending on the length of the recording, the normalizing denominator (total power) can be with or without the *ULF* component (sum of power includes *ULF*, *VLF*, *LF* and *HF* for 24 recordings, and includes *VLF*, *LF* and *HF* for short-term recordings [20]). Even though the normalized spectral HRV indices can be interpreted in the same manner as the absolute values, the general guideline recommends reporting the low-frequency and high-frequency components in both absolute (e.g., *LF*, *HF*) and normalized forms (e.g., *LFn*, *HFn*) to present a complete picture of the power distribution [18].

## 8. Time-Frequency Methods of Analysis

The spectral analysis algorithms used in frequency-domain approaches can only compute the power spectrum density (PSD) over the entire recording, failing to account for the temporal variation of the frequency components. To address this limitation, tools to simultaneously capture both time and frequency changes (i.e., a time-frequency approach, TF), have been proposed [37,63]. By evaluating the frequency spectrum over short moving windows, TF methods quantify the instantaneous PSD at any given time [19]. The TF methods include, among others, linear approaches such as the short-time Fourier transform (*STFT*), the wavelet transform, and quadratic approaches such as the Wigner–Ville distribution (*WVD*).

As it allows for the integration over both the time and frequency domain, the TF HRV indices usually include the maximum and minimum energy of the time windows in HR signal, the standard deviation between energy of time windows, the total energy of signal in different frequency bands (e.g., energy of *VLF*, *LF* or *HF*), and the corresponding average energy of each band (derived by dividing total energy by length of band). While the TF approaches are believed to be more informative since they provide a means to quantitatively assess the changes in frequency components across time, they are not as widely used as other domains. One current limiting factor is that some of them are computationally expensive, while others, being easier to compute, can hardly provide high accuracy in both time and frequency domains [64,65]. The appeal of TF approaches could be greatly enhanced by addressing these technical challenges, together with future works that investigate the interpretations of relevant TF features extracted from the time-varying spectrum.

## 9. Non-Linear Dynamics

Recent research has emphasized non-linear interactions of physiological systems underlying cardiovascular regulation [66]. Hence, HRV indices derived from non-linear methods were proposed to adequately characterize the dynamical properties of HR that other methods are unable to [18].

## 10. Poincaré Plot

The Poincaré plot, whose principles were derived from nonlinear dynamics theory, is one of the most common methods to measure non-linear HRV [67]. It corresponds to a scatterplot of each NN interval plotted against its corresponding preceding interval, which approximates the cardiac system’s evolution [68]. The points are dispersed around the identity line and converge into an ellipsoid configuration. Points above the line represent HR decelerations (NN intervals that are longer than preceding ones) and points below the line of identity indicate HR accelerations [69].

Qualitatively, the Poincaré plot’s shape provides a visual summary of the heart’s behavior [67]. From the plot, outliers such as premature heartbeats and other technical artefacts can be easily removed, something which cannot be done with spectral and most time-domain analyses. On a quantitative level, indices extracted from the plot correspond to time-domain indices. *SD1* is the standard deviation of points perpendicular to the identity line and is equivalent to *RMSSD* [48], describing short-term NN variability. *SD2* is the standard deviation of points parallel to the line of identity and is equivalent to *SDNN*, representing long-term NN variability. The former reflects short-term vagal activity whereas the latter represents sympathetic modulation. A third index, *SD1/SD2*, describes the ratio of short to long-term variations in NN interval fluctuations and reflects sympathovagal balance [70]. *SD1/SD2* has been shown to additionally capture complexity in HR patterns (e.g., healthy heart displays a “comet” shaped Poincaré plot while cardiac abnormalities display atypical “fan” or “complex” shapes [71,72,73]). However, *SD1/SD2* seems to be sensitive to the time lag used, with some recent research recommending a lag of five or six heartbeats [72,74]. Nevertheless, there is no consensus on the optimal time lag yet, which itself could be subject to inter-individual differences.

## 11. Entropy

Entropy-based methods find their roots in information theory in which they are conceptualized as measures of complexity (or “unpredictability”) of a signal. While they originated from fields of thermodynamics [75], their applications have expanded towards the quantification of complexity in physiological systems and have been used widely as a diagnostic tool in biomedicine. The underlying concept of entropy is that it quantifies repetitions of patterns in a signal, with greater entropy indicating higher randomness and unpredictability (and thus, complexity) and lower entropy values implying that the cardiac system is predictable, (i.e., periodic and does not produce much new information) [76]. Common entropy-based methods include approximate entropy (*ApEn*) [77], sample entropy (*SampEn*) [78], and multiscale entropy (*MSE*) [79]. *SampEn* was built on *ApEn* [78] and addresses the latter’s tendency to overestimate the amount of regularity in the signal [75,80,81], being in turn more consistent in measuring complexity across different time series lengths [67]. However, another important difference is that *ApEn* computes regularity based on short time sequences whereas *SampEn* considers the entire time series [75]. Consequently, *ApEn* might be better suited to capture short-term HRV changes than *SampEn*, which is more often used as a global index of variability [76].

However, as higher irregularity is not always equivalent to higher complexity [82], these traditional measures have been shown to falsely quantify random HRV changes in certain pathological conditions as being complex [79,83]. In order to estimate complexity from irregularity statistics more accurately, Costa et al. [83] suggested instead examining irregularity in multiple spatio-temporal scales where *SampEn* is estimated on more than one scale, giving rise to a new entropy-based method called *MSE*. Compared to single-scale entropy estimates, *MSE* has been able to accurately account for lower HRV observed in different cardiac abnormalities, such as atrial fibrillation where there are random outputs in HRV [83]. As *MSE* formulation does not allow for reliable computation of entropy in short time windows, new methods such as refined MSE (*RMSE*) [84] and refined composite MSE (*RCMSE*) [85] have been introduced, though few studies have tested them on HRV [86,87].

The entropy analysis of HRV is an actively evolving field, with more novel approaches being proposed, each claiming to confer certain advantages over more traditional approaches. For example, new measures such as fuzzy entropy (*FuzzEn*) [88] and fuzzy measure entropy (*FuzzyMEn*) [89] that were developed to overcome the poor statistical stability of *ApEn* and *SampEn* [90,91] have been gaining traction. While a comparison of clinical validity across these entropy measures has been demonstrated in a few studies (e.g., [89,92,93]), the choice of entropy type and the appropriate parameters for entropy calculation is not a straightforward process and investigation into how these factors influence HRV interpretations is still ongoing [94].

## 12. Fractal Measures

Fractal methods are more recently developed methods to assess noisy and nonstationary time series. In the context of HRV, fractal methods measure the fluctuations of the cardiac system across multiple time scales, where lower self-similarity/self-affinity (i.e., similar structural details at different scales of visualization) may imply a randomly structured, less adaptable and flexible system.

Correlation dimension (*CD*) is a widely used measure of fractal dimension (an index of how patterns change with the scale at which it is measured). In the case of HRV, *CD* measures the number of dynamic variables needed to define the underlying system of the HR time series, where a higher *CD* reflects greater complexity in HRV. Other algorithms for computing fractal dimension also exist, such as the box-counting dimension, and the algorithm by Katz [95] and Higuchi [96], but *CD* is the most popular by far [67].

In recent years, the most extensively used method in measuring HRV complexity is detrended fluctuation analysis (*DFA*) [97]. *DFA* is a measure of the correlations between successive NN intervals at different time scales, and is used to analyze recordings spanning several hours. Its computation disregards nonstationarities in the cardiac time series and hence any spurious correlations due to artifacts of nonstationarity behavior (e.g., trends due to external stimuli) are avoided, an advantage conferred over other fractal methods [98,99]. *DFA* eventually extracts both long-term correlations and short-term correlations, with anywhere between 4 and 16 heartbeats for the former, and 16 and 64 heartbeats for the latter [98]. The short-term correlations correspond to the baroreceptor reflex whereas the long-term correlations measure regulatory mechanisms in the cardiac system [20]. See Table 1 for a summary of all indices.

## 13. Respiratory Sinus Arrhythmia (RSA)

RSA refers to the fluctuations of HR at the respiratory frequency, or in other words, the effect of respiratory-gated PNS activity on the heart. As RSA commonly occurs at high-frequency heart rate fluctuations, most studies quantify RSA using the *HF* component of the spectral measures [100]. Other methods to quantify RSA include the Porges–Bohrer method (*PB*) [101] and the peak-to-trough algorithm (*P2T*) [102,103]. *PB* extracts RSA by removing variances in the HR time series that are not attributable to spontaneous breathing frequencies, and has high accuracy only when there is a low signal-to-noise ratio in the data. On the other hand, *P2T* derives RSA by finding the difference between the shortest heart period during inhalation and the longest heart period during exhalation per single breath.

Even though the three different RSA indices are highly correlated [102], *PB* has been shown to be less susceptible to the non-stationarity in the HR signal and the influence of respiration than the *HF* component or *P2T* metric [23]. While both *HF* and *P2T* may work well in the absence of slow components and trends [23], *PB* has been proposed to be the most sensitive to vagal activity.

## 14. HRV in Psychological Research

As mentioned above, the different HRV indices are believed to reflect different physiological mechanisms. It is therefore unsurprising that they are associated with distinct psychological constructs, such as cognitive abilities, neural processes, personality traits and psychopathological disorders.

## 15. HRV and Self-Control

Based on the neurovisceral integration model [104,105], the neural circuitry implicated in the effective regulation of psychological processes (i.e., emotional and cognitive self-regulation) are also related to the regulation of cardiac autonomic activity. The model has gained ample support over the past decades as mounting evidence emerges, showing a significant relationship between HRV and prefrontal cortex activity [105,106,107,108]. This is in line with studies investigating the relationship between HRV and executive functioning, showing that participants with higher resting HRV (particularly short-term changes) performed better at different executive tasks, such as sustained attention [109,110], working memory [111,112] inhibition [106,113,114,115], and cognitive flexibility [16,116,117]. Additionally, Hansen et al. [111] also highlighted the executive-specific advantage of higher vagally-mediated resting HRV which was not observed in non-executive tasks based on simple reaction time.

Autonomic flexibility (i.e., the ability of ANS to modify bodily states), indexed by HRV, is crucial for adjusting between high and low arousal states, and has thus been related to emotional regulation. In studies that investigated emotional responses to stress, individuals with higher resting RSA had lower perceived (negative) emotional arousal and made greater use of adaptive coping strategies [118,119]. Similarly, individuals with higher *HF* experience more positive moods, in particular cheerfulness and calmness, and this relationship was mediated by emotion regulation abilities (e.g., ability to engage in situational planning, refocusing, and reappraisal) [120]. This parallels findings reporting that anxious participants experienced a greater decrease in *RCMSE* during stress exposure [86]. This is in line with neuroimaging findings highlighting the relationship between HRV and the activity of brain regions involved in threat perception and saliency detection such as the amygdala, media prefrontal cortex [121]. Although the evidence in favor of a link between cognitive and affective dysregulation and reduced HRV is robust, the exact physiopsychological mechanisms underlying this association remain to be uncovered [122].

While most of the literature focuses on the role of resting HRV in cognitive control and emotional regulation, another level of analysis of HRV, the phasic HRV, has been less examined but is also of great interest for health and well-being. Resting or tonic HRV is measured at the state of resting, whereas phasic HRV is measured during the state of stress where changes in HRV, relative to baseline, represent the reactivity of the cardiac system to different stimuli. While the advantages of higher tonic HRV have been repeatedly shown in the literature [121], the desirability of phasic HRV changes (increase versus decrease phasic HRV) is context-dependent [123]. Higher phasic HRV suppression (–the autonomic stress response to prepare for fight-or-flight situations [124] has been linked to higher levels of tonic HRV and phasic HRV post-stress, which are indicators of faster recovery [125,126,127,128]. Therefore, researchers have highlighted its potential role as a protective factor against stress [123,129,130]. However, when the stressor requires the engagement of executive functioning [131], higher phasic HRV enhancement is preferred as it indicates greater attentional resources and self-regulatory effort [132,133,134]. Interestingly, by looking at participants’ performance under different levels of perceptual loads, Park et al. [123] observed that individuals with higher tonic HRV can exert regulatory effort (presence of phasic HRV enhancement) and avoid making autonomic stress response (absence of phasic HRV suppression) even under high loads where the task could be a potential stressor. In contrast, individuals with lower tonic HRV experienced autonomic stress response (phasic HRV enhancement) even in low loads condition, suggesting a tendency to appraise even trivial challenges as stressors. Given the significant interaction between tonic and phasic HRV and its potential to provide insight on individual differences in regulatory processes, future studies should investigate not only HRV at rest but also HRV reactivity (changes between rest and event) and HRV recovery (changes between event and post-event) [135].

## 16. HRV and Psychopathology

Research focusing on the relationship between HRV and mood disorders suggests that, in general, a lower vagal tone (and thus, a lower HRV) is associated with negative outcomes [136]. Short-term HRV indices such has *RMSSD*, *SDANN*, *HF* and non-linear measures such as *MSE* were found to be significantly reduced in patients with depression [14,137,138] and negatively correlated with symptom severity [139]. Short-term HRV indices are also lower (and correlated with symptoms’ severity [140]) in anxiety disorders, including generalized anxiety disorder (GAD), social anxiety disorder, panic disorder and post-traumatic stress disorder [11,12,14,17,124,141,142,143,144,145,146,147] Additionally, depression with comorbid anxiety has been shown to be associated with greater reduction in HRV as compared to depression alone [15]. In this context, lower HRV has been associated with changes in the sympathovagal balance in favor of a sympathetic predominance, which may subsequently increase cardiovascular risk in these pathologies [146]. Bipolar disorder is also associated with a decreased HRV across a variety of HRV indices, including *RMSSD*, *pNN50*, *SDNN*, *HF*, *LF*, and *SampEn* [148,149,150,151]. A similar pattern would be expected with borderline personality disorder, given the shared physiopsychological mechanisms with bipolar disorder [152]. However, the few studies currently report mixed findings (e.g., a higher *HF* in BPD patients [153]) vs. lower RSA [154]). Research also suggests a lower HRV with reduced *RMSSD*, *HF*, *ApEn* and *SampEn* in patients suffering from schizophrenia [149,155,156,157,158], and a correlation with symptoms such as apathy, withdrawal and delusions [159], suggesting a relationship between psychotic symptoms and cardiac autonomic function [149,160].

HRV has been presented as a useful clinical marker of psychopathology, in which it represents the combined effect of parasympathetic inhibition as well as excessive sympathetic activation, reflecting a sub-optimal adaptive response to external and internal changes. Although the specific pathophysiological mechanisms underlying the associations between psychiatric conditions and HRV remain poorly understood, the majority of studies observed significant relationships between the different pathologies and parasympathetic parameters (such as *HF* and *RMSSD*, that are particularly sensitive to short-term HR changes). This has led researchers to propose that a restrained vagal activity could be a transdiagnostic biomarker across psychiatric disorders [17,155,161,162]. This hypothesis is further supported by the observation of worsening parasympathetic tone during the chronic course of psychiatric conditions [155,163], and parallel improvements in parasympathetic parameters as symptom severity reduces over the course of treatment [14,145,164,165].

## 17. HRV as a Therapeutic Target

With the progressive establishment of HRV as a convenient peripheral (and correlational) index of self-regulatory abilities [104,166], its usage in applied settings grew. Recently, the psychophysiological cardiac coherence model has been introduced to promote training to enhance HRV. This model emphasizes the importance of having reciprocal interactions and coherence between the regulatory systems to maximize the adaptability to environmental demands [167,168]. Cardiac coherence refers to a stable state of increased synchrony between the rhythms of different oscillatory systems, typically HR, respiration and blood pressure. In practice, this coherent state is reflected by a more ordered and sine wave-like HRV waveform, oscillating at a frequency approximating 0.1 Hz, the frequency at which the phase synchrony of HR and respiration occurs [169,170].

According to the model, cardiac coherence can be induced by breathing at a steady pace approximating 0.1 Hz (i.e., six breaths per minute). Such deliberate slow-paced breathing (also referred to as “deep breathing,” [170]) maximizes both baroreflex sensitivity and RSA on HR, giving rise to a high-amplitude oscillation of HR [171,172]. According to [172], regular breathing exercising techniques help to continuously stimulate and exercise autonomic reflexes, especially the baroreflex. In fact, deep breathing has been shown to produce significant improvements in cognitive and emotional functioning [167] and different dimensions of health and well-being, such as pain [173], stress and anxiety [174,175], depression [176,177,178], post-traumatic stress disorder [179], or attention-deficit hyperactivity disorder [180]. Different mechanisms have been proposed to explain the therapeutic effects of HRV-focused biofeedback [181], but further investigations are needed to understand, clarify and validate its potential clinical utility.

## 18. HRV in Practice: A Tutorial Using Python

Over the years, HRV has become a convenient and reliable marker to detect ANS and cardiovascular abnormalities. However, due to the complex mathematical models and theories of signal processing involved, there is a practical barrier for users without the prior technical knowledge, such as many psychology students and researchers, to effectively understand, interpret, produce and reproduce HRV indices. In the following, we will present a step-by-step guide to obtain HRV indices using NeuroKit2 [182]), an open-source Python package that is designed to be friendly to both novice and advanced programmers. Instructions on how to get Python and NeuroKit2 up and running can be found at https://neurokit2.readthedocs.io/en/latest/installation.html (accessed on 3 May 2021). Note that other open-source software such as *HeartPy* [183], *pyHRV* [184], *hrv* [185] and GUI-based software such as *KUBIOS©* [186], and *BIOPAC Acqknowledge software* (https://www.biopac.com) (accessed on 3 May 2021) are other alternatives for HRV analysis.

## 19. Data Preprocessing

Cardiac activity data stored on the disk can be loaded using functions appropriate to the format (e.g., *read_bitalino()* or *read_acqknowledge()*). For the purposes of this tutorial, we will download and use an example dataset available in NeuroKit2, which consists of recordings of four participants during 8 min of resting state. Their cardiac activity (ECG) and respiration (RSP) signals were recorded and downsampled to 200 Hz for the purpose of this demonstration (note that higher sampling rates are recommended to maximize the precision of NN intervals).

The data from the first participant is then extracted, stored into an object, and two of its variables (the ECG and RSP signals) are passed to a preprocessing function that will clean the signals and, importantly, find the location of R peaks and respiration cycles. Details about the options and algorithms used are available in the package’s documentation.

*# Load data*
import neurokit2 as nk
data_all = nk.data(‘https://raw.githubusercontent.com/neuropsychology/NeuroKit/dev/data/bio_resting_8min_200hz.json’)
data_participant = data_all[“S01”] *# Access the data of the first participant*

*# Clean signal*
clean_signals, info = nk.bio_process(ecg=data_participant[“ECG”],
 rsp=data_participant[“RSP”],
 sampling_rate=200)

*# Show the location of the fist 5 R-peaks*
info[“ECG_R_Peaks”][0:5]

## array([165, 345, 506, 672, 838])


The *bio_process()* function allows simultaneous processing of different signals using signal-specific pipelines. For instance, for ECG signals, the steps involve minimizing noise in the data, removing artifacts due to environmental or biological sources, as well as extracting all potential peaks (e.g., R-peaks) in the signal before removing peaks that are not of interest (i.e., ectopic peaks).

NeuroKit2 offers flexibility in choosing different processing methods. For this tutorial, we will use the *bio_process()* function which allows for less customization of the pipeline and relies on sensible default parameters for performing signal-specific operations (i.e., cleaning, peak extraction, rate computation). Alternatively, users can customize their own processing pipeline following the tutorial available on the *documentation of NeuroKit2*.

## 20. HRV Analysis

HRV indices of different domains can then be obtained using specific functions such as *hrv_time(), hrv_frequency(), hrv_nonlinear()* and *hrv_rsa()*, to which the list of R-peaks location, the sampling rate (200 Hz in this example), and respiration phases (specifically for *hrv_rsa()*) have to be provided. However, all the available indices can also be computed at once using *hrv()*, alongside a figure combining useful diagnostic plots (such as the power spectral density and the Poincaré plot) when the *show* argument is set to *True*. Table 2 shows an example of the indices output and Figure 1 shows an example of the diagnostic plot.

*# Compute all HRV indices and show plots*

nk.hrv(info, sampling_rate=200, show=True)


## 21. Multiple Participants Analysis

As the function(s) to compute HRV indices for a participant return a single row dataframe, the process can easily scale up for studies that contain multiple participants by, essentially, binding all the rows together into one dataframe. In the example below, we will demonstrate how to loop through participants, compute HRV indices at each turn and append the data. For more complex or tailored analyses, it is within such loops that additional participant-specific operations can be inserted (e.g., apply specific data preprocessing steps or extract other types of information).

*# Load data and create list of participants*
data_all = nk.data(‘https://raw.githubusercontent.com/neuropsychology/NeuroKit/dev/data/bio_resting_8min_200hz.json’)

list_participants = [“S01”, “S02”, “S03”, “S04”]

*# Loop through participants’ names* 
**for** participant **in** list_participants:

 *# Get data of current participant*
 data_participant = data_all[participant]

 *# Clean signal*
 clean_signals, info = nk.bio_process(ecg=data_participant[“ECG”],
 rsp=data_participant[“RSP”],
 sampling_rate=200)

 *# Compute HRV indices*
 results = nk.hrv(info, sampling_rate=200, show=False)

 *# Add “Participant” column*
 results[“Participant”] = participant

 *# If participant is not the first, then append the data* 
 **if** participant == “S01”:
 results_all = results
 **else:**
 results_all = results_all.append(results)


## 22. Conclusions

Throughthe above literature review related to HRV and its applications in psychology and neuroscience, we demonstrated how HRV analysis can be efficiently performed using NeuroKit2 by computing the indices, creating useful diagnostic plots, as well as by aggregating the indices across multiple subjects. We hope that helping to provide psychologists with a comprehensive list of HRV indices will help to shift the focus from physiological preprocessing and programming technicalities towards the interpretation of the results involving these indices as well as the critical discussion of their limitations.

Given the amount of existing HRV indices, in addition to the lack of clarity regarding their associated physiological mechanisms, psychophysiological researchers might find it difficult to pinpoint which ones to focus on. Time-domain indices have a long history of usage in HRV research, with RMSSD being arguably the most robust measure of vagal tone [44,135]. While these statistical indices are quick to compute, it is crucial to carefully preprocess the HR signal before performing HRV analysis, as their accuracy can be easily affected by outliers and artefacts. In cases of noisy signal, geometrical approaches such as the HTI might be a more viable alternative. Frequency-domain indices may provide greater precision in quantifying autonomic control mechanisms. Among the spectral measures, the relationship between HF and vagal tone has been the most well-established. However, given its sensitivity to processing parameters chosen, it is recommended that HF is interpreted in conjunction with other time-domain indices that are thought to similarly reflect vagal tone (e.g., RMSSD, pNN50, see [135]). Controlled breathing protocols can also be used to minimize variable respiratory influences on HF across different populations (e.g., athletes have slower respiratory rates, children breathe faster, [44,187]). Additionally, although the use of LF/HF has also been quite popular amongst researchers, definitive conclusions about sympathovagal balance based solely on this parameter are ill-advised due to the unclear sources of LF.

While both time- and frequency-domain indices are correlated with non-linear indices, the latter are more reflective of the general complexity of HR signal rather than any specific underlying physiological mechanism [188]. However, theoretical and methodological concerns related to their usage have been raised. Firstly, these indices are very sensitive to the parameters chosen (e.g., window length), with unclear guidelines, rules, or standards for their (optimal) selection. Secondly, while some studies attest to the clinical utility of non-linear HRV dynamics when applied longitudinally (i.e., ability to differentiate progression of cardiovascular diseases), it is not yet clear why certain non-linear methods are better at distinguishing between pathologies. Apart from methods covered in this paper, other methods such as largest Lyapunov exponent (LLE), Hurst exponent (HE) and symbolic dynamics have also gained wide acceptance for assessing various complex systems, but their applications in HRV are substantially less popular than the aforementioned non-linear indices [67]. All in all, it is recommended to treat these indices as complementary measures to traditional linear indices [135,188].

Finally, other considerations influence the choice and interpretation of HRV indices, such as recording length, quality of the recorded signal, breathing methods and participant variables (e.g., age, gender, cardio-related medication, etc.) [18,189]. To facilitate the comparisons of HRV-related results across populations and studies, past studies have attempted to quantify the reference values for HRV measures of different recording lengths (e.g., see [190,191] for long-term recordings; [192,193] for short-term recordings), of different populations (e.g., [194] for healthy school-aged children; [193] for healthy adults) and/or across different demographic characteristics (e.g., [195,196]). Related to this, researchers have provided a comprehensive list of practical recommendations [188] to improve the use of standardized protocols for HRV research. All in all, the focus of the current paper is aimed at providing an overview of HRV indices, their usage in psychological research, and a practical demonstration on how to obtain these indices using NeuroKit2.

## Figures and Tables

**Figure 1 sensors-21-03998-f001:**
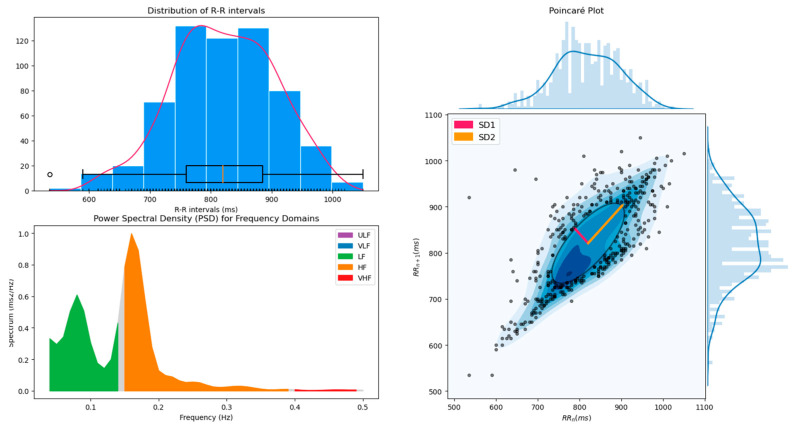
Output of the *hrv()* function showing the distribution of NN intervals, the power spectrum density (PSD) and the Poincaré plot, that can be used as a visual diagnostic tool for HRV.

**Table 1 sensors-21-03998-t001:** A summary of HRV indices according to their respective analysis domains.

Analysis Domain	Sub-Domain	Acronym	Description
Time-Domain	Deviation-based approach	SDNN	Standard deviation of NN intervals
		SDANN	Standard deviation of average NN intervals for each 5 min segment
		SDNN Index	Mean of the standard deviations of NN intervals in 5 min segments
	Difference-based approach	SDSD	Standard deviation of successive NN interval differences
		RMSSD	Root mean square of successive NN interval differences
		pNN20	Proportion of successive NN interval differences larger than 20 ms
		pNN50	Proportion of successive NN interval differences larger than 50 ms
	Geometric approach	HTI	Integral of the density of the NN interval histogram divided by its height
		TTIN	Baseline width of the RR interval histogram
Frequency-domain	Absolute Power	ULF	Power spectrum in the frequency range of ≤0.003 Hz
		VLF	Power spectrum in the frequency range of 0.0033–0.04 Hz
		LF	Power spectrum in the frequency range of 0.04–0.15 Hz
		HF	Power spectrum in the frequency range of 0.15–0.4 Hz
	Normalized/Relative Power	LnHF	Natural logarithm of HF
		HFn	Normalized HF
		LFn	Normalized LF
		LF/HF	Ratio of LF to HF power
Time-frequency domain	Linear approach	STFT	Power spectrum estimation using short-term Fourier transform
		WT	Power spectrum estimation using Wavelet transform
	Quadratic approach	WVD	Power spectrum estimation using Wigner–-Ville distribution
		SWVD	Power spectrum estimation using Smoothed Wigner–Ville distribution
Non-linear domain	Poincaré Plot	SD1	Poincaré plot standard deviation perpendicular the line of identity
		SD2	Poincaré plot standard deviation along the line of identity
		SD1/SD2	Ratio of SD1 to SD2
	Entropy	ApEn	Estimation of complexity using approximate entropy
		SampEn	Estimation of complexity using sample entropy
		MSE	Estimation of complexity using multiscale entropy
	Fractal Dimensions	DFA	Estimation of signal fluctuations using detrended fluctuation analysis
		CD	Estimation of minimum number of variables to define a dynamic model

**Table 2 sensors-21-03998-t002:** The *hrv()* function returns a dataframe containing all the indices as columns.

HRV_RMSSD	HRV_MeanNN	…	HRV_DFA	HRV_CorrDim
61.74	820.75	…	0.66	1.40

## Data Availability

Not applicable.

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
