# Peer review of "Heart Rate Variability in Psychology: A Review of HRV Indices and an Analysis Tutorial"

_sensors, 2021, doi:10.3390/s21123998_

Round 1

Reviewer 1 Report

This is a broad and informative review regarding the different variables used in the study of Heart Rate Variability. It will be of great interest to beginners in the subject, due to the availability of the extensive literature consulted. It is noteworthy, however, the lack of a study that is quite relevant in this context and was not mentioned. This is the study by Bilman GE published in Front Physiol. 2013 Feb 20; 4:26. doi: 10.3389 / fphys.2013.00026. eCollection. In this study, the LF component of the frequency domain is discussed, in the sense that it is not actually a representative of the sympathetic component as mentioned by the authors, but that it may even have more parasympathetic than sympathetic components. As the present article is a review article and intends to be a tutorial, it is essential that this approach taken by Gilman be added. Also noteworthy is the absence of a table containing the reference values, for each of the variables, cited in the literature. We recommend adding a table and a query to the following references, which should also be added:

Nunan D, Sandercock GRH, Brodie DA. A Quantitative Systematic Review of Normal Values ​​for Short-Term Heart Rate Variability in Healthy Adults. PACE, 33: 1407–1417; 2010

Sammito S, Böckelmann I. Reference values ​​for time and frequency-domain heart rate variability measures. Heart Rhythm, Vol 13, No 6, June 2016

Author Response

This is a broad and informative review regarding the different variables used in the study of Heart Rate Variability. It will be of great interest to beginners in the subject, due to the availability of the extensive literature consulted.

It is noteworthy, however, the lack of a study that is quite relevant in this context and was not mentioned. This is the study by Bilman GE published in Front Physiol. 2013 Feb 20; 4:26. doi: 10.3389 / fphys.2013.00026. eCollection. In this study, the LF component of the frequency domain is discussed, in the sense that it is not actually a representative of the sympathetic component as mentioned by the authors, but that it may even have more parasympathetic than sympathetic components. As the present article is a review article and intends to be a tutorial, it is essential that this approach taken by Gilman be added.

Thank you for highlighting this important point. We do agree with the reviewer that the interpretation of LF power as an index of sympathetic activities is misleading and not accurate. We have made the following amendment to the following paragraph to further highlight this issue [line 200]:

Particularly, the interpretation of LF power as an index of sympathetic activities has been repeatedly challenged. In contrast to the assumption that LF power dominantly representing the sympathetic component, research has shown that it is modulated by both ANS branches as well as baroreceptor activities (Acharya et al., 2006; Billman, 2013; Force, 1996; Goldstein et al., 2011).

Also noteworthy is the absence of a table containing the reference values, for each of the variables, cited in the literature. We recommend adding a table and a query to the following references, which should also be added:

Nunan D, Sandercock GRH, Brodie DA. A Quantitative Systematic Review of Normal Values ​​for Short-Term Heart Rate Variability in Healthy Adults. PACE, 33: 1407–1417; 2010

Sammito S, Böckelmann I. Reference values ​​for time and frequency-domain heart rate variability measures. Heart Rhythm, Vol 13, No 6, June 2016

We agree with the reviewer that HRV normative data is important information, particularly in the context of clinical applications and clinical research. We did not venture into detail about the reference values because firstly, its practical usefulness is still somewhat limited in the context of general psychological research and secondly, there remains a lack of expansive studies that establish reliable normative data for individuals of different groups (e.g., age, gender, habitual physical activity levels). Though there are several attempts to quantify the reference values (e.g., Force, 1996; Nunan et al., 2010; Sammito et al., 2016; GÄ…sioret al., 2018), there are discrepancies observed across these studies and they are potentially due to the differences in measurement standards (e.g., recording lengths, recording tools, sampling frequencies) and/or data analysis pipelines (e.g., analysis algorithms and software). Additionally, the majority of past studies focused on a few common time- and frequency- domain values and thus normative data for more recent measurements such as time-frequency domain or non-linear analysis remains lacking. 

However, as we aim to provide a thorough review of HRV indices, we agree that it is important to inform our readers about the works that have been done to establish the different reference values. Following the reviewer’s comment, we have added the following to the manuscript [line 652]:

To facilitate the comparisons of HRV-related results across populations and studies, past studies have attempted to quantify the reference values for HRV measures of different recording lengths (e.g., see Aeschbacher et al., 2017; Tegegne et al., 2020 for short-term recordings; Umetani et al., 1998; Nunan et al., 2010 for long-term recordings), of different populations (e.g., Gasior et al., 2018 for healthy school-aged children; Dantas et al., 2018 for healthy adults) and/or across different demographic characteristics (e.g., Berg et al., 2018; Sammito & Bockelmann, 2016).

Reviewer 2 Report

This is an excellent primer and review for psychologists who wish to use HRV. It includes non-linear analyses, very useful for those only familiar with time domain and frequency domain measurements.

The authors refer to Catai et al and Force for information concerning the choice and interpretation of HRV indices, including technical issues and participant variables. A follow up paper addressing these issues in detail would be welcomed. It would also be helpful to discuss ECG vs PPG data, and how NeuroKit compares with Kubios.

Author Response

This is an excellent primer and review for psychologists who wish to use HRV. It includes non-linear analyses, very useful for those only familiar with time domain and frequency domain measurements.

The authors refer to Catai et al and Force for information concerning the choice and interpretation of HRV indices, including technical issues and participant variables. A follow up paper addressing these issues in detail would be welcomed.

It would also be helpful to discuss ECG vs PPG data, and how NeuroKit compares with Kubios.

Thank you for your encouraging comments on our manuscript. A follow-up paper to explore and address the challenges pertaining to the choice and the interpretation of HRV indices would indeed be something that we would like to pursue.

A general comparison between NeuroKit and other physiological processing software (such as Kubios) has been included in the original publication of NeuroKit (Makowski et al., 2021) and a more in-depth comparison can be found in NeuroKit's documentation (https://neurokit2.readthedocs.io/en/latest/examples/hrv.html). Thus, in this manuscript, we prefer to focus on the description of the HRV indices per se and their computation using non-commercial and open-source software (which Kubios is not), which supports the movement towards open-science. However, we do agree that users should be informed about other alternatives for HRV analysis and hence, we have added the following sentence [line 517]:

Note that other open-source software such as HeartPy (van Gent et al., 2019), pyHRV (Gomes et al., 2019), hrv (Bartels & Pecanha,2020), and GUI-based software such as KUBIOS© (Tarvainen et al.,2014), and BIOPAC Acqknowledge software (https://www.biopac.com) are other alternatives for HRV analysis.